# Prescribed hormonal contraceptive use trends in the Estonian Biobank: A longitudinal observational study

Jelisaveta Džigurski[1]*, Märt Möls[1,2], Kristi Läll[1], Hannah Currant[3,4], Mall Eltermaa[5], Estonian Biobank Research Team[1¶], Reedik Mägi[1], Lili Milani[1], Triin Laisk[1]*

1 Estonian Genome Centre, Institute of Genomics, University of Tartu, Tartu, Estonia, 2 Institute of Mathematics and Statistics, University of Tartu, Tartu, Estonia, 3 Nuffield Department of Women's and Reproductive Health, University of Oxford, Oxford, United Kingdom, 4 Big Data Institute, University of Oxford, Oxford, United Kingdom, 5 Institute of Biomedicine and Translational Medicine, University of Tartu, Tartu, Estonia

¶ Membership of the Estonian Biobank Research Team is provided in the Acknowledgments.
* jelisaveta.dzigurski@ut.ee (JD); triin.laisk@ut.ee (TL)

## Abstract

### Background

Hormonal contraceptives (HCs) are widely used and have well-documented population-level statistics. Previous studies with short follow-ups have focussed on individual HC use and side effects. However, the same aspects over longer periods, HC formulation switching, and the impact of genetic factors on HC side effects remain understudied due to the limited availability of suitable datasets. We investigated whether the Estonian Biobank (EstBB) is suitable for studying genetic risk for HC side effects.

### Methods and findings

This is a longitudinal descriptive study combining prescribed HC purchase data collected from 2004 to 2022 with genetic and health data from 73,071 female EstBB HC users aged 15–55 at the time of purchase. HC usage was defined by the Anatomical Therapeutic Chemical (ATC) codes G02B, G03A, and G03HB01. Methods included calculating age-stratified annual user prevalence, inferring usage periods from purchases, assessing formulation switching, identifying the International Classification of Diseases, Tenth Revision (ICD-10)-based side effect-related diagnoses and thromboembolism risk factors, and assessing carrier status for Factor V Leiden (FVL, rs6025) and prothrombin G20210A (PTM, rs1799963) genetic variants as proof-of-concept. Over 19 years, 20 HC formulations with five administration routes (oral pills, transdermal patches, vaginal rings, subdermal implants, intrauterine devices) were used. In the EstBB, combined HCs were the most commonly used among users aged 15–29, while progestin-only HC use increased with age and over time, comparable to

**Data availability statement:** The analyses carried out in this study included individual-level data, which cannot be made publicly available due to legal and ethical restrictions. To access the data, the approval must be obtained from the Scientific Advisory Committee of the Estonian Biobank and the Estonian Committee on Bioethics and Human Research. The inquiry for data should be sent via e-mail to releases@ut.ee. For more details, please see the Data Access section at https://genomics.ut.ee/en/content/estonian-biobank#dataaccess. The source code used in this study is publicly available from GitHub (https://github.com/alter-echo11/hc_use_estbb) and archived on Zenodo (https://doi.org/10.5281/zenodo.19615879).

**Funding:** TL and JD are funded by the Estonian Research Council (https://etag.ee/en/) grants TK214 and PSG776. MM is funded by the Estonian Research Council (https://etag.ee/en/) grant PRG1911. RM and KL are funded by the Estonian Research Council (https://etag.ee/en/) grants TK214 and PRG1911. LM is funded by the Estonian Research Council (https://etag.ee/en/) grant PRG2625. HC is funded by the Wellcome Trust (https://wellcome.org/) 318918/Z/24/Z and the National Institutes of Health (NIH: https://www.nih.gov) 1P50HD104224-01. The funders had no role in study design, data collection and analysis, decision to publish, or preparation of the manuscript.

**Competing interests:** We have read the journal's policy and the authors of this manuscript have the following competing interests: KL has participated as an analyst in a research grant awarded to the Institute of Genomics from Geneto OÜ. All other authors have declared that they have no competing interests.

**Abbreviations:** ATC, Anatomical Therapeutic Chemical; ATE, arterial thromboembolism; BMI, body mass index; CHCs, combined hormonal contraceptives; CI, confidence interval; COCs, combined oral contraceptives; DRSP, drospirenone; EHIF, Estonian Health Insurance Fund; EHRs, electronic health records; EMA, European Medicines Agency; EstBB, Estonian Biobank; FVL, Factor V Leiden; HCs, hormonal contraceptives; ICD-10, International Classification of Diseases, Tenth Revision; IQR, interquartile range; LNG, levonorgestrel; OR, odds ratio; PCOS, polycystic ovary syndrome;

the Estonian population. Overall, 64.2% ($n = 46{,}920$) of users switched formulations at least once, with 17.7% ($n = 12{,}929$) being rapid switchers. Side effect-related diagnoses were observed in 23.1% ($n = 2{,}982$) of rapid switchers, with excessive/irregular menstrual bleeding being the most common. Genetic analysis revealed that 5.3% ($n = 3{,}886$) of users carried at least one variant previously associated with increased thrombosis risk (3.5% ($n = 2{,}556$) carried FVL only, 1.8% ($n = 1{,}276$) PTM only, and 0.07% ($n = 54$) both). Carriers of thrombosis-associated variants had a significantly higher percentage of thrombosis (6.5%) than non-carriers (4.2%; OR = 1.61, 95% CI [1.40, 1.84], $p < 0.001$). The study is limited by the use of administrative medication purchase records, which may not reflect actual HC use. Additionally, diagnoses identified near the HC usage period may not represent true side effects.

## Conclusions

This study provides insights into real-world HC usage with longitudinal, individual-level detail that is not available in population-level statistics. We show that EstBB has a robust dataset for studying the impact of genetic factors on HC side effects and disease risk. The identified HC user profiles offer a framework for genetic analyses of HC rapid switching and discontinuation. Our approach can be replicated in other biobanks to validate findings across populations.

---

### Author summary

#### Why was this study done?

- A better understanding of how genetics interacts with hormonal contraceptives (HCs) and affects the risk of side effects could improve HC prescribing, yet suitable datasets for studying this have been lacking.

- The Estonian Biobank (EstBB) is a volunteer-based population biobank with genetic, health-related and medication purchase data, but the representativeness of the sample regarding HC usage needs to be investigated.

#### What did the researchers do and find?

- We reconstructed EstBB participants' HC usage periods from the HC purchase data and linked them to the users' genetic data and medical diagnoses.

- We found that the EstBB HC usage trends were comparable to those of the Estonian population.

- We identified a subgroup of HC users who switched rapidly between HC formulations, with side effect-related diagnoses near the switch, suggesting our approach can indicate women who are potentially experiencing side effects.

POCs, progestin-only contraceptives; POPs, progestin-only pills; PR, prevalence ratio; PTM, prothrombin G20210A; RECORD, Reporting of Studies Conducted using Observational Routinely Collected Data; SD, standard deviation; STROBE, Strengthening the Reporting of Observational Studies in Epidemiology; VTE, venous thromboembolism.

- We found that women who use HCs and carry genetic variants associated with blood clot risk had more blood clot diagnoses than non-carriers.

## What do these findings mean?

- This study highlights the potential of the EstBB dataset for HC research and offers a reproducible framework.

- Population-based biobanks with genetic and health-related data, such as the EstBB, offer a valuable resource for studies investigating pharmacogenetic aspects of HCs.

- This study relies on records of HC purchases accumulated over a long period, and from these records alone, we could not confirm whether HCs were used as purchased. Diagnoses identified near the HC usage period may not represent true side effects. Also, because HC use varies across countries, our findings may not apply to women residing outside Estonia.

### Introduction

Hormonal contraceptives (HCs) are used by approximately 300 million women worldwide [1]. HCs are mainly used to support women's reproductive goals and treat symptoms associated with gynaecological conditions, such as polycystic ovary syndrome (PCOS) [2], endometriosis [3], and heavy menstrual bleeding [4]. Various HC formulations with different administration routes have been designed to meet women's needs and overall health. Generally, the active components are either a combination of progestin and estrogen (combined hormonal contraceptives (CHCs)) or progestin alone (progestin-only contraceptives (POCs)). Each hormonal method is associated with potential side effects and health risks [5], which may lead to either discontinuation or switching of a medication [6,7]. Most notably, CHCs are associated with increased risk of venous thromboembolism (VTE) and arterial thromboembolism (ATE) [8,9]. Despite potential risks, the European Medicines Agency (EMA) has stated that the risk of VTE with CHCs is low and that the benefits outweigh any potential risks; only a woman's individual thromboembolism risk should be evaluated before prescribing CHCs [10]. Similar prescription recommendations exist for individuals experiencing certain conditions, such as migraine [11,12].

The availability of specific HC formulations and prescription practice vary across countries, reflecting differences in healthcare systems and sociocultural norms [1]. A study published in 2021 found that during 2005–2019, the use of POCs increased in Estonia, while CHC use declined [13]. At the same time, it was found that there was a considerable number of CHC users with a history of thrombosis or multiple documented risk factors for thrombosis. Both VTE and ATE have several risk factors that should be evaluated before prescribing HCs. These include family history of thromboembolism, age, smoking status, and underlying medical conditions, such as hypertension, cardiovascular disease, and others [14]. The use of antipsychotics or antidepressants can also increase thrombosis risk [15,16], although the results

for antidepressants have been conflicting and could be due to drug-drug interactions [17,18]. Moreover, VTE has well-established genetic risk factors, such as rare antithrombin, protein C and protein S deficiencies, and more common gain-of-function variants in the coagulation factor V (*F5*) and coagulation factor II (*F2*) genes (found in approximately 1%–5% of individuals of European ancestry) [19,20]. The association of these common genetic variants with ATE remains uncertain, with most studies showing modest or small effects [21,22]. Current World Health Organization guidelines do not support routine screening for thrombophilia before prescribing HCs [11]. However, a recent study using United Kingdom (UK) Biobank data suggested that evaluating the full spectrum of genetic risk for thrombosis among oral HC users may improve risk stratification [23]. While this study highlighted the value of large population-based biobanks for studying genetic risk factors of HC side effects, due to the age profile of the participants the results applied mainly to second-generation oral HCs and relied on self-reported use.

The Estonian Biobank (EstBB) is a volunteer-based population biobank with 212,000 participants (~140,000 females, 43% currently aged under 50), for whom genotype and health-related data are available [24]. The electronic health records (EHRs) of EstBB participants are regularly linked to the Estonian Health Insurance Fund (EHIF) and the Estonian Health Information System, providing nearly complete coverage of diagnoses and prescriptions for all participants, dating back to the early 2000s when this type of information was digitalised. This rich dataset has formed the basis for studies exploring the genetic architecture of polygenic traits [25,26], phenotypic comorbidities [27], and pharmacogenetics [28,29]. Given the large number of reproductive-aged women, EstBB is an ideal research setting for studying HC usage and its pharmacogenetic aspects. However, previous studies have shown a "healthy participant bias" in the widely used volunteer-based biobanks [30,31]. Currently, we lack an overview of how the EstBB data compares to the Estonian population regarding HC use trends and risk factor profiles.

In this study, we describe prescribed HC use trends from 2004 to 2022 and user profiles among EstBB female participants. This unique cohort and longitudinal analysis provide a comprehensive overview of prescription practices, individual HC use trajectories and switching across the reproductive life span. We compare these trends with Estonian national statistics to assess how representative the EstBB cohort is of the Estonian population's HC use. Additionally, we evaluate the presence of thromboembolism risk factors among HC users, including both clinical risk factors and genetic variants (Factor V Leiden (FVL) and prothrombin G20210A (PTM)). Overall, we aim to investigate whether the EstBB dataset is suitable for future genetic studies of HC-related side effects and disease risk.

## Methods

This is a descriptive study without a predefined hypothesis, aiming to comprehensively characterise observed HC use in the EstBB and evaluate its potential for future genetic analyses of HC-related side effects and disease risk. The study is reported according to the Strengthening the Reporting of Observational Studies in Epidemiology (STROBE) guideline (Table A in S1 Checklist) and the Reporting of Studies Conducted using Observational Routinely Collected Data (RECORD) guideline (Table B in S1 Checklist) [32,33]. The study did not have a prospective protocol or analysis plan. In response to peer review suggestions, one additional analysis was completed: evaluation of thromboembolism risk factors among POC users and CHC users stratified by estrogen dose [34].

### Study population and participants

EstBB is a volunteer-based population biobank with data from more than 210,000 individuals who were 18 years or older at recruitment, of whom 65% were female [24]. The cohort represents approximately 20% of the total adult population of Estonia ($n_{total\_adult}$ = 1,077,308 in 2025) [35]. Nearly all participants (99.5%) are insured by the EHIF [24], which covers 95% of the total Estonian population [36], suggesting the potential for selection bias due to insurance status is minimal. The comprehensive description of the cohort has been published previously [24]. Health-related data are linked from the Estonian Health Information System, EHIF, Tartu University Hospital Database, and many others. The information about

medical diagnoses dates to 2004 and is stored according to the ICD-10 Version:2016 system. The EstBB 206K data freeze—with follow-up until the end of 2022—was used in this study.

All female EstBB participants were included in the study, but the focus was on those with at least one prescribed HC purchase record between 2004 and 2022 and aged 15–55 at the time of purchase. Basic descriptive characteristics of HC users, such as the age of the first observed HC usage period, the age at the start of each HC usage period, and the body mass index (BMI) closest to the start of each usage period, were used for analysis. BMI values originated from self-reported data or have been extracted from EHRs and were categorised by Centers for Disease Control and Prevention criteria: (a) underweight (BMI < 18.5), (b) healthy weight (18.5 ≤ BMI < 25.0), (c) overweight (25.0 ≤ BMI < 30.0), and (d) obese (BMI ≥ 30.0).

### Hormonal contraceptive usage

In Estonia, all HCs except for emergency contraceptives require a prescription. Data on HC prescriptions and purchases originates from the EHIF, which subsidises medical expenses for people covered by national health insurance. The purchase invoice records the purchase of prescribed HC in a pharmacy. The linking procedure has been previously described in Leitsalu and colleagues [37]. Specifically, our dataset includes the purchase invoices from 2004 to 2022. For analysis, we used the Anatomical Therapeutic Chemical (ATC) Classification System code and the active substance name of the prescribed and purchased drug, the purchased dosage, the package contents, and the purchase date.

HCs were defined as purchased prescriptions of ATC level 3 G02B and G03A, and ATC level 5 G03HB01 (Fig 1). There was no information about emergency contraceptives (ATC level 4 G03AD), as a prescription is not needed for their purchase in Estonia. All entries where prescribed and purchased package ATC codes did not match were excluded from the analysis ($n_{purchases} = 30,930$ (1.8%)). In the following filtering step, all entries of short-acting HCs where the purchased dosage was less than 0.3 or greater than 6 were excluded ($n_{purchases} = 653$ (0.04%)). This had a negligible impact on prevalence estimates and analyses, and ensured the dataset was free from mistype errors. In the Estonian healthcare system, buying less than a third of a HC package is not possible, and gynaecologists or midwives prescribe, per visit, a maximum of three prescriptions, each for two short-acting HC packages (i.e., for a six-month coverage period). Finally, purchases for which the individual's age at purchase was outside the 15–55 range ($n_{purchases} = 1,306$ (0.1%)) and individuals for whom BMI values closest to the HC usage start date were outside the range of 14.0–43.0 ($n_{individuals} = 851$ (1.1%)) were excluded.

The purchasing data was converted into HC usage periods. Specifically, the purchased dosage and package contents for each HC method were used to calculate the total number of covered days per purchase, assuming that the contraceptive was used consistently and according to guidelines. The default number of covered days per purchase for pills, transdermal patches and vaginal rings was set to 28 or 35 (latter only for G03AC03), for implants to 1,095 days (corresponding to three years), while the coverage days for intrauterine devices (IUDs) depended on the formulation dosage (1,095, 1,835 or 2,190 days corresponding to 3, 5 or 6 years for 13.5, 19.5 mg or 52 mg levonorgestrel-releasing IUD (LNG IUD), respectively). Inferred HC usage periods were additionally censored with ICD-10 pregnancy codes (O00–O99, *excl.* O85–O92 (complications predominantly related to the puerperium)) that occurred within the original coverage period. For example, if a pregnancy code appeared within an inferred HC usage period, we ended that period at the pregnancy diagnosis date. We did not backdate before the pregnancy date, as pregnancy-related diagnoses vary widely in gestational timing. This approach was applied across all HC types to avoid differential bias. Regular purchases (<90 days of pause between two purchase dates) of the same HC were considered one usage period since the fecundity is generally restored in two to three months after discontinuation [38,39]. Furthermore, for HC formulations with intermittent dosing, the drug-free days were also considered as part of the usage period. Any purchase of a different HC formulation (different ATC level 5 code) would mark the start of a new usage period. There was no differentiation between brands, dosages, or regimes for the same HC formulation.

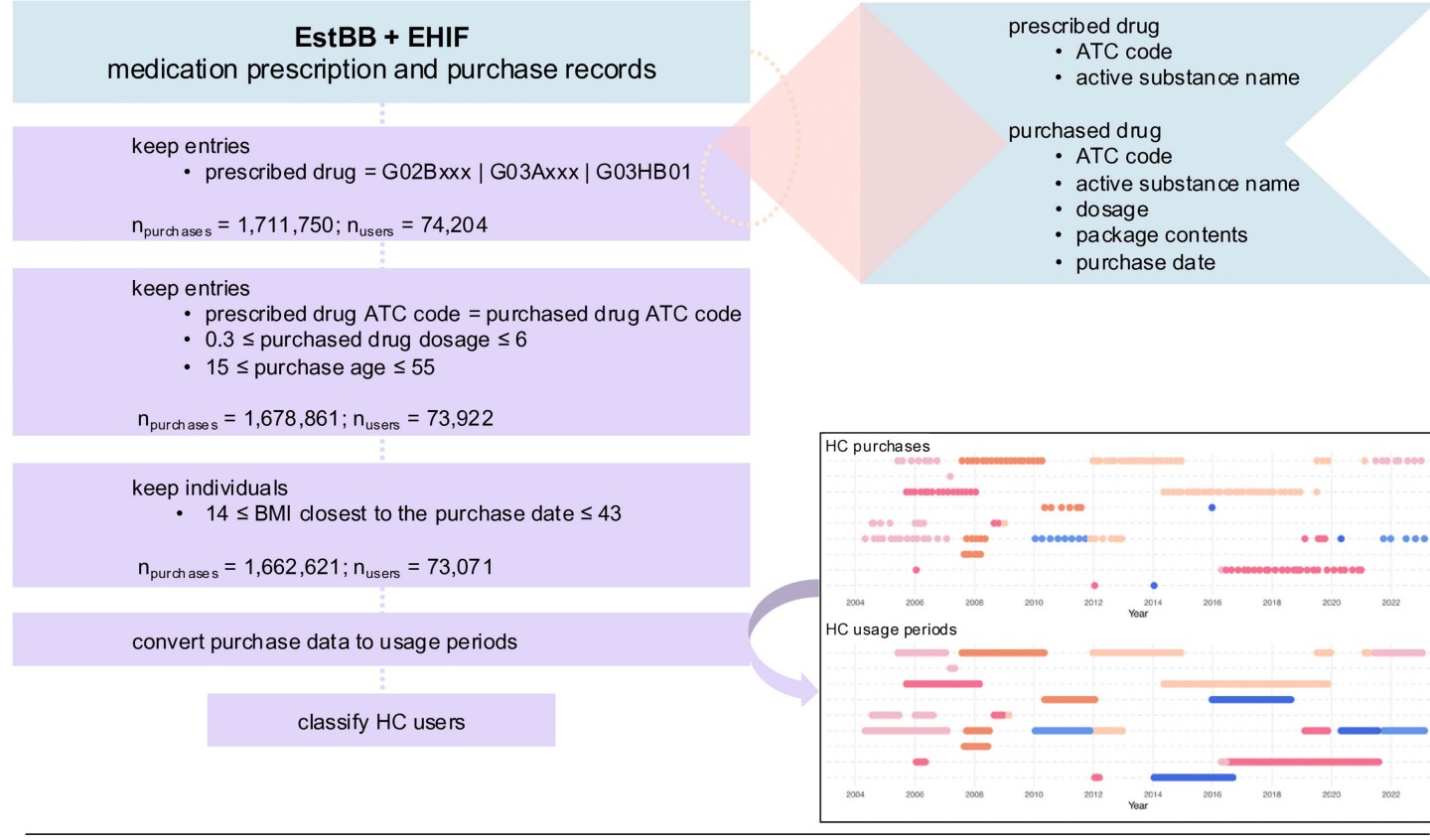

**Abbreviations**: EstBB – Estonian Biobank; EHIF – Estonian Health Insurance Fund; ATC – Anatomical Therapeutic Chemical Classification System; HC – Hormonal Contraceptive

**Fig 1. Flowchart of input data preparation for the construction of hormonal contraceptive (HC) usage periods and classification of HC user types using linked Estonian Biobank (EstBB) and Estonian Health Insurance Fund (EHIF) medication prescription and purchase data.** Several filtering steps were applied to identify eligible purchases and users: (i) selecting purchases of relevant Anatomical Therapeutic Chemical (ATC) codes (G02Bxxx, G03Axxx, G03HB01); (ii) matching prescribed and purchased ATC codes, excluding mistyped dosages, and restricting purchases to the relevant purchase age range; and (iii) restricting to individuals with body mass index (BMI) values within the eligible range, recorded closest to the purchase date. The final number of purchases ($n_{purchases}$) and users ($n_{users}$) after each step is shown. The final HC purchase data were converted into HC usage periods, which were used to classify HC users. The manually defined data points are for illustrative purposes and are not real observations.

To answer different research questions, HCs were grouped into ATC levels 4 or 5 accordingly and categorised by hormonal composition, route of administration, duration of action and contraceptive method type. Combined oral contraceptives (COCs) were classified by pharmacological properties and progestin generations as: (a) anti-androgenic (cyproterone/ethinylestradiol, chlormadinone/ethinylestradiol), (b) second-generation (levonorgestrel/ethinylestradiol), (c) third-generation (desogestrel/ethinylestradiol, gestodene/ethinylestradiol, norgestimate/ethinylestradiol, norelgestromin/ethinylestradiol), (d) fourth-generation (drospirenone/ethinylestradiol, dienogest/ethinylestradiol), and (e) estradiol preparations (nomegestrol/estradiol, dienogest/estradiol valerate). While grouping COCs by generation is informal [40], it is a common practice in the literature, where different COC generations are associated with different side effects profiles, especially thrombosis risk levels [41,42]. However, as it also varies across studies, we explicitly define our grouping criteria for clarity.

Finally, all HC users were classified as broad switchers, rapid switchers, non-switchers and rapid discontinuers following these criteria: (a) broad switchers were individuals who used two or more different HC formulations during the study

observation period, (b) rapid switchers were individuals who had short gap (≤90 days) between two different formulation usage periods and short (<90 days) pre-switch formulation usage period, (c) non-switchers were individuals who had at least one long (>180 days) usage period and used only one formulation during the study observation period, and (d) rapid discontinuers were individuals who had only one short (<90 days) usage period and used only one formulation during the study observation period. Rapid discontinuers were additionally verified if they had any recorded diagnoses after the first purchase date, as evidence of not moving out of Estonia (which could otherwise result in missing purchases), and the purchase was made more than 90 days before the end of the study period (to rule out artefacts created by the end of the observation period). Overall, broad switchers are a wider class, with rapid switchers being its subset, while the remaining classes are mutually exclusive of each other and switchers (S1 Appendix). While the terminology was adopted from exist-ing research [43,44], the thresholds were defined based on the clinical guidelines recommending an initial three-month 'trial' period and the Estonian prescription system.

### Longitudinal diagnosis history and thromboembolism risk factors

To identify clinical diagnoses before, during and after HC use, EHRs with ~4,500 ICD-10 three-character category and four-character subcategory diagnosis codes were used. The illustration of the strategy is shown in Fig A of S2 Appendix. First, the records of clinical diagnoses in the three-month window/90 days preceding the first-time-observed HC usage period start date were used to identify potential non-contraceptive reasons for HC initiation (Table A in S3 Appendix).

Second, when evaluating diagnoses during the HC usage period and up to three months after the usage period end date to identify diagnoses related to potential side effects and possible reasons for HC discontinuation, we only consid-ered first-time-ever diagnoses to avoid misinterpretation from pre-existing conditions. The same approach was used for diagnoses during the gaps between two different HC usage periods of rapid switchers. Our list of the curated selection of diagnoses related to potential side effects, based on side effects listed in patient information leaflets (Table A in S3 Appendix).

Third, the prevalence of VTE/ATE was evaluated across the entire study period. VTE and ATE were defined using ICD-10 diagnosis codes (Table A in S3 Appendix).

Expanding on the approach of Kurvits and colleagues [13], in the subset of women who purchased HCs in 2022, thromboembolism risk factors were assessed in a two-year period ending on the first purchase date in 2022 (Fig B in S2 Appendix). All thromboembolism risk factors were divided into three classes: (a) health conditions, (b) medication use, and (c) genetic risk factors. Health conditions were defined using the ICD-10 codes (Table A in S3 Appendix) [13,45]. Previous thrombotic events, family history of thrombotic events in first-degree relatives (parent–offspring or full siblings), and age > 35 in 2022 were considered additional risk factors. The familial relationship between participants was inferred centrally for the EstBB. It is estimated based on coefficients calculated from the genetic data (kinship inference and identical-by-descent segment inference for close relatives) using the KING software, which implements a robust relation-ship inference algorithm [46].

The concomitant use of other medications associated with an increased risk of thromboembolism during the HC usage period was identified using purchasing data of ATC codes H02 (corticosteroids), N05A (antipsychotics), and N06A (antide-pressants) [13].

Additionally, the imputed genotype data were assessed for the presence of two known VTE-associated variants (FVL (rs6025, MAF = 0.02) in the *F5* gene and PTM (rs1799963, MAF = 0.01) in the *F2* gene. The genotyping and imputation procedure of the EstBB cohort has been described previously [24]. Briefly, all samples have been genotyped centrally using the Global Screening Array (Illumina) and imputed using a population-specific reference panel of 2,695 whole-genome sequencing samples [47]. Imputation INFO scores for both variants were >0.96, confirming the good quality of the genotypes.

## Statistical analyses

The categorical data were summarised using frequency calculations, and the continuous data were summarised by calculating the mean, standard deviation (SD), median and interquartile range (IQR [Q1, Q3]) where relevant. The annual use of HCs was calculated by summing up all active users in a given year. Annual HC users were stratified into five age groups (15–19, 20–29, 30–39, 40–49, and 50–55) using age at purchase, calculated by subtracting the birth year from the purchase year. The age at the start/end of the HC usage period was calculated similarly to the age at purchase. The first-ever observed HC usage period during the study was considered the initiation period, and users were referred to as initiators. Annual prevalence was calculated by dividing the number of active users in the specific age group in the given year by the total number of women in the EstBB of that age group in that year. The proportion of each HC type for each calendar year and age group was calculated as the total number of covered days by HC type divided by the total number of covered days by all HCs in that year and age group. The prevalence ratio (PR) with 95% confidence interval (CI) was used to compare thromboembolism risk factor prevalence between HC groups, for consistency, as these provide more interpretable effect estimates for common outcomes (>10%). For other categorical associations, the odds ratio (OR) with 95% CI was reported. Fisher's exact test was used to assess statistical significance for all comparisons between categorical variables. The significance threshold was set to $p < 0.05$, except for risk factor comparisons, where Bonferroni correction was applied to account for multiple testing across 15 factors ($p < 0.003$). The data were analysed and visualised using the following packages: *R base, data.table (v1.15.4)* [48], *dplyr (v1.1.4)* [49], *tidyr (v1.2.0)* [50], *stringr (v1.5.1)* [51], *lubridate (v1.9.3)* [52], *formattable (v0.2.1)* [53], *epitools (v0.5.10.1)* [54], *ggplot2 (v3.5.1)* [55], *ggalluvial (v0.12.5)* [56], *viridis (v0.6.5)* [57], *scales (v1.3.0)* [58] and *patchwork (v1.2.0)* [59] in the R Statistical Software v4.1.3 [60].

## Ethics approval and consent to participate

The ethical approval 1.1–12/624 for the study was granted by the Estonian Committee on Bioethics and Human Research (Estonian Ministry of Social Affairs), using data according to release application number 6–7/GI/16011 from the Estonian Biobank. All participants have signed a broad written informed consent form.

## Results

### General use trends

The study cohort included 132,731 women, and we focussed on 73,071 who had purchased HC at least once between the ages of 15 and 55. HC user characteristics are summarised in Table 1. Women contributed 1,040,298 person-years of follow-up (from the first purchase date), with an average follow-up of 14.2 years per person. The average age at the first HC purchase was 27.9 (SD = 9.2) years, with the highest number of initiators at the age of 19 ($n_{users} = 4,842$). The BMI distribution at the first purchase date was comparable to that reported for the Estonian female population of similar age [61].

Across 19 years, 20 different HC formulations with five distinct routes of administration (intrauterine device, intravaginal ring, oral pill, transdermal patch and subdermal implant) were used. The most widely used HCs were COCs containing gestodene/ethinylestradiol (GSD/EE, $n_{users} = 27,651$) and drospirenone/ethinylestradiol (DRSP/EE, $n_{users} = 26,030$), levonorgestrel-releasing IUD (LNG IUD, $n_{users} = 24,465$), desogestrel-only pill (DSG, $n_{users} = 16,557$) and CHC ring containing etonogestrel/ethinylestradiol (ENG/EE, $n_{users} = 13,376$). Detailed information about all purchased HCs across the study period is presented in Table 2.

The demographic structure of our study cohort has shifted notably over the years, reflected in a decline in the proportion of individuals aged (constructed age, see Methods) 15–29, from 40.6% in 2004 to 15.3% in 2022 (Fig 2A, Table B in S3 Appendix). Accordingly, the annual prevalence of CHC users was much higher than that of POC users, but has decreased with time (Fig 2B, Table C in S3 Appendix), while the prevalence of POC users steadily increased. Although our estimated user prevalence was systematically higher than that reported by Kurvits and colleagues [13], who estimated the

**Table 1. Demographic characteristics of Estonian Biobank (EstBB) women (*N* = 73,071) who purchased a prescribed hormonal contraceptive (HC) at least once between ages 15 and 55 in the period from 2004 to 2022.**

| Characteristic | Value |
|---|---|
| EstBB enrolment | |
| Age at enrolment, mean (SD) | 35.6 (10.6) |
| Age at enrolment, median (IQR) | 35 (27–43) |
| Age at enrolment<30 years, *n* (%) | 23,967 (32.8) |
| First HC purchase | |
| Age at first purchase, mean (SD) | 27.9 (9.2) |
| Age at first purchase, median (IQR) | 20 (25–35) |
| Modal age at first purchase, years (*n*) | 19 (4,842) |
| First purchase before EstBB enrolment, *n* (%) | 65,219 (89.2) |
| First purchase after EstBB enrolment, *n* (%) | 7,852 (10.8) |
| BMI at first purchase | |
| Mean (SD) | 24.3 (4.6) |
| Median (IQR) | 23.1 (20.9–26.4) |
| Underweight (<18.5), *n* (%) | 3,159 (4.3) |
| Healthy weight (18.5–24.9), *n* (%) | 45,136 (61.8) |
| Overweight (25.0–29.9), *n* (%) | 16,093 (22.0) |
| Obese (≥30.0), *n* (%) | 8,683 (11.9) |

BMI range reflects predefined inclusion criteria (see Methods). Abbreviations: EstBB, Estonian Biobank; SD, standard deviation; IQR, interquartile range; *n*, number; HC, hormonal contraceptive; BMI, body mass index.

annual prevalence of HC users aged 15–49 in Estonia from 2005 to 2019, the pattern of yearly changes was the same. Specifically, the annual prevalence of POC users in the EstBB increased across all age groups (15–19: from 0.3% in 2004 to 10.9% in 2022, 20–29: from 3.9% in 2004 to 17.2% in 2022, 30–39 from 3.2% in 2004 to 21.3% in 2022, 40–49: from 0.9% in 2004 to 27.6% in 2022 and 50–55: from 0.03% in 2004 to 18.0% in 2022). The annual prevalence of CHC users in the adolescent and the oldest age groups of EstBB increased (15–19: from 17.0% in 2004 to 38.2% in 2022, and 50–55: from 1.0% in 2004 to 2.0% in 2022), while for the other age groups it decreased (20–29 from 39.4% in 2004 to 26.6% in 2022, 30–39 from 21.5% in 2004 to 14.8% in 2022, and 40–49: from 9.6% in 2004 to 8.5% in 2022). Similar trends were observed by Kurvits and colleagues [13], except that our dataset showed an increase in CHC user prevalence in groups 15–19 and 20–29 from 2015 onward, and their study lacked information for the 50–55 group.

The changes in the distribution of total covered days (i.e., active use) by HC type among women from different age groups throughout the study observation period are presented in Fig 2C and Table D in S3 Appendix. While generally there was a continuous decrease in anti-androgenic, 2nd and 3rd generation COC use, the proportion of 4th generation COC use was increasing, then remained steady in more recent years. An exception to this trend was observed in the 15–19 and 20–29 age groups, where the proportion of 2nd generation COC use began increasing again from 2015 (Fig 2C and Table D in S3 Appendix). The use of estradiol COCs and HC implants began in 2010 and 2014, respectively. The proportions of different HC types varied slightly between years, reflecting the availability of specific formulations, as some multiphasic COCs and progestin-only pills (POPs), such as gestodene CHC (GSD/EE(s)), levonorgestrel-only pill (LNG) and drospirenone-only pill (DRSP), were only available on the market and used for 2–4 years.

While levonorgestrel-releasing IUDs (LNG-IUD) were increasingly used and primarily by older women (accounting for 89.4% of active HC use in the 50–55 age group in 2022), CHC dominated among younger and middle-aged women,

**Table 2. Overview of hormonal contraceptives (HCs) used by Estonian Biobank participants from 2004 to 2022.**

| HC ATC Code | HC Abbreviation Name | HC Formulation Name | Main Hormonal Components | Route of Administration | Duration of Action | HC Type | Number of Individuals | Number of Purchases |
|---|---|---|---|---|---|---|---|---|
| G03AA10 | GSD/EE | gestodene and ethinylestradiol | progestin + estrogen | oral | short-acting | COC 3rd generation | 27,651 | 370,347 |
| G03AA12 | DRSP/EE | drospirenone and ethinylestradiol | progestin + estrogen | oral | short-acting | COC 4th generation | 26,030 | 425,757 |
| G02BA03 | LNG IUD | plastic IUD with progestogen | progestin | intrauterine | long-acting | IUD with progestogen | 24,465 | 37,431 |
| G03AC09 | DSG | desogestrel | progestin | oral | short-acting | Progestin-only pill | 16,557 | 79,335 |
| G02BB01 | CHC RING | vaginal ring with progestogen and estrogen | progestin + estrogen | intravaginal | short-acting | CHC ring | 13,376 | 160,743 |
| G03AA13 | NGMN/EE | norelgestromin and ethinylestradiol | progestin + estrogen | transdermal | short-acting | CHC patch | 13,182 | 125,341 |
| G03AA16 | DNG/EE | dienogest and ethinylestradiol | progestin + estrogen | oral | short-acting | COC 4th generation | 12,765 | 178,867 |
| G03AA09 | DSG/EE | desogestrel and ethinylestradiol | progestin + estrogen | oral | short-acting | COC 3rd generation | 11,229 | 118,347 |
| G03HB01 | CPA/EE | cyproterone and ethinylestradiol | progestin + estrogen | oral | short-acting | Anti-androgenic COC | 5,068 | 52,309 |
| G03AC03 | LNG | levonorgestrel | progestin | oral | short-acting | Progestin-only pill | 4,318 | 13,758 |
| G03AA15 | CMA/EE | chlormadinone and ethinylestradiol | progestin + estrogen | oral | short-acting | Anti-androgenic COC | 3,063 | 21,879 |
| G03AA07 | LNG/EE | levonorgestrel and ethinylestradiol | progestin + estrogen | oral | short-acting | COC 2nd generation | 2,763 | 17,433 |
| G03AB03 | LNG/EE(s) | levonorgestrel and ethinylestradiol | progestin + estrogen | oral | short-acting | COC 2nd generation | 2,350 | 33,275 |
| G03AC08 | ENG | etonogestrel | progestin | subdermal | long-acting | Implant | 1,576 | 1,979 |
| G03AA11 | NGM/EE | norgestimate and ethinylestradiol | progestin + estrogen | oral | short-acting | COC 3rd generation | 972 | 12,215 |
| G03AB08 | DNG/E2V | dienogest and estradiol valerate | progestin + estrogen | oral | short-acting | Estradiol COC | 574 | 5,336 |
| G03AC10 | DRSP | drospirenone | progestin | oral | short-acting | Progestin-only pill | 400 | 1,232 |
| G03AB06 | GSD/EE(s) | gestodene and ethinylestradiol | progestin + estrogen | oral | short-acting | COC 3rd generation | 397 | 1,944 |
| G03AA14 | NOMAC/E2 | nomegestrol and estradiol | progestin + estrogen | oral | short-acting | Estradiol COC | 328 | 2,017 |
| G03AB05 | DSG/EE(s) | desogestrel and ethinylestradiol | progestin + estrogen | oral | short-acting | COC 3rd generation | 309 | 2,562 |
| G03AA10(p) | GSD/EE(p) | gestodene and ethinylestradiol | progestin + estrogen | transdermal | short-acting | CHC patch | 261 | 514 |

The table is sorted by the total number of individuals using HCs. The "(s)" indicates combined hormonal contraceptive pill with sequential preparations, and "(p)" patch.

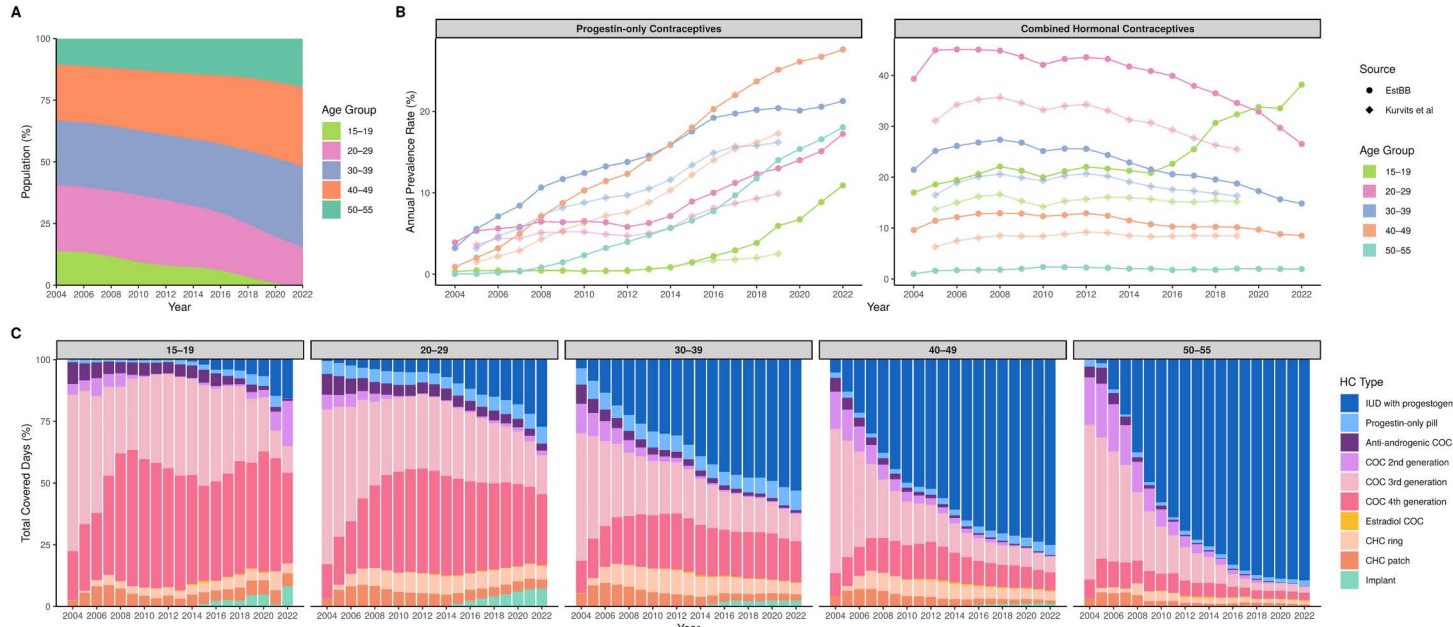

**Fig 2. Overview of hormonal contraceptive (HC) use trends from 2004 through 2022. A**: Age dynamics in the Estonian Biobank (EstBB) study sample (HC users and non-users combined) during the study period. **B**: Annual prevalence (%) of HC users in different age groups across studied years in EstBB (darker colour) and Kurvits and colleagues [13] (lighter colour), shown separately for two main HC classes, progestin-only hormonal contraceptives and combined hormonal contraceptives. **C**: Distribution of total covered days by HC type across studied years, stratified by age groups (15–19, 20–29, 30–39, 40–49, 50–55). Users using more than one HC type within the same calendar year contribute to each relevant group. Abbreviations: IUD, intrauterine device; COC, combined oral contraceptive; CHC, combined hormonal contraceptive.

especially in the early 2000s and 2010s, with the highest usage of 3rd and 4th generation COCs. On the other hand, POPs accounted for the highest proportion of active HC use among the 20–29 and 30–39 age groups (6.9% and 8.0% in 2022, respectively), with these percentages remaining stable over the years. Estradiol COCs were the least prevalent type of HCs and had minimal variation in their share of active HC use across the studied period (up to 0.9% of active HC use across all age groups throughout the years). CHC rings and patches were almost equally prevalent across all age groups and displayed a declining trend (Fig 2C). Etonogestrel-releasing (ENG) implants were more common over the years among adolescents aged 15–19 (8.2% in 2022) and women aged 20–29 (7.2% in 2022), whereas among women in older age groups, the proportions remained steadily low.

### First observed hormonal contraceptive usage period

The first purchased prescriptions in 83.4% of women were CHCs, notably 36.1% were 3rd generation COCs and 24.9% were 4th generation COCs, irrespective of BMI category (S4 Appendix). However, each increase in the year of the first purchase was associated with lower percentages of 3rd generation COCs (decreasing from 55.6% of initiators in 2004 to 14.8% in 2022) and higher percentages of LNG-IUDs (increasing from 1.7% of initiators in 2004 to 36.0% in 2022). The 4th generation COCs peaked in 2011 with 38.7% of all initiators choosing this method. POPs were most commonly used by the 30–39 age group (11.4%) and by women from the obese category (9.4%), while LNG-IUDs were chosen by 39.5% of women from the 50–55 age group. Estradiol COC and ENG implant were the least used options for HC initiation (0.2% and 0.3%, respectively). The HC formulation abbreviations and their definitions are presented in Box 1.

### Box 1. Hormonal contraceptive abbreviations used in the study.

| | |
|---|---|
| LNG IUD | Levonorgestrel intrauterine device |
| LNG | Levonorgestrel |
| DSG | Desogestrel |
| DRSP | Drospirenone |
| CPA | Cyproterone acetate |
| EE | Ethinylestradiol |
| CMA | Chlormadinone acetate |
| GSD | Gestodene |
| NGM | Norgestimate |
| DNG | Dienogest |
| NOMAC | Nomegestrol acetate |
| E2 | Estradiol |
| E2V | Estradiol valerate |
| NGMN | Norelgestromin |
| ENG | Etonogestrel |

The first contraceptive period length of specific HC formulations varied from less than a month (<30 days) to more than three years (>1,095 days), with a median of approximately 5.5 months (168 days) as the most common for the majority of HCs, except for long-acting methods, such as LNG-IUD and ENG implant, and multiphasic CHCs, such as LNG/EE(s) and GSD/EE(s) with higher medians (2,190, 1,027, 384, and 279 days, respectively, Fig 3). Some usage period lengths were underestimated when the first purchase was made near the end of the observation period or when the woman continued using the method past the age of 55, which exceeded the study age limits.

Looking into the ICD-10 three-character category diagnosis codes recorded up to three months before the first HC usage period start date, we found 1,233 different codes, where 53.1% ($n_{users}$ = 38,820) of HC initiators had a Z30 diagnosis code related to contraceptive management, and 11.2% ($n_{users}$ = 8,151) did not have any record of the medical condition. Additionally, 4.8% ($n_{users}$ = 3,531) had heavy menstrual bleeding (N92), 4.0% ($n_{users}$ = 2,922) medical abortion (O04), 2.4% ($n_{users}$ = 1,744) dysmenorrhoea (N94), 1.6% ($n_{users}$ = 1,202) acne (L70), and 0.7% ($n_{users}$ = 472) PCOS (E28.2), which were from the list of curated diagnoses we considered as a potential reason for starting HCs.

Previous studies have shown changes in prescribing and purchasing trends during the Coronavirus Disease 2019 (COVID-19) pandemic, driven by lockdowns and limited access to healthcare services [62]. We investigated the impact of the COVID-19 pandemic and pandemic-induced lockdowns on HC initiation, comparing 2019 (pre-COVID-19 period) to 2020 and 2021 (COVID-19 periods). Overall, up to 55.4% of first-ever initiators were consistently from the 20–29 age group, and up to 50.0% of initiators with a history of prior HC use were from the 30–39 age group. Our longitudinal cohort's yearly number of HC users gradually declined from 2019 to 2021. While the population-weighted average rate of first-ever initiators decreased from 1.3 to 1.1 to 0.8 during this period, other initiators decreased from 9.4 to 8.6 to 8.3, which likely also reflects the ageing dynamics of our study cohort. Furthermore, the overall monthly purchase trends differed notably in 2020 (Fig A in S5 Appendix), with March 2020 20% above the grand mean and April 2020 20% below it. Specifically, March had higher purchase numbers of CHC pills, patches, rings, and POPs than March 2019, with the peak on March 13th (Fig D in S5 Appendix). Conversely, LNG-IUDs and implants had their lowest record numbers in March and April 2020. Following June 2020, only 2nd-generation COCs and LNG-IUDs had a slight increase in purchases compared to the corresponding periods in 2019. Also, compared to 2021, only 2nd-generation COCs, estradiol COCs, implants, LNG-IUDs, and POPs returned to numbers similar to those recorded in the pre-COVID-19 period.

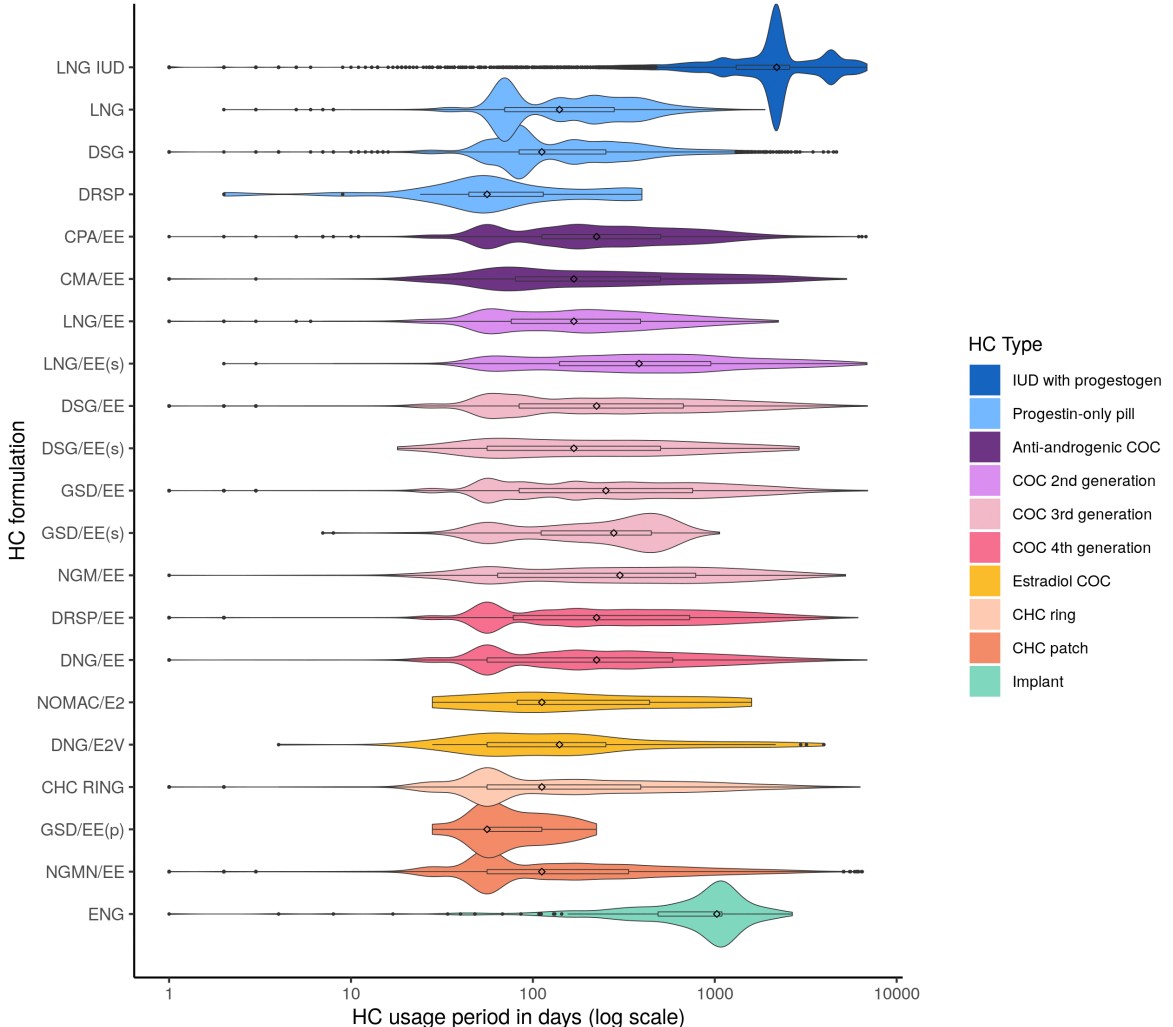

**Fig 3. Length of first observed hormonal contraceptive (HC) usage period for different HC formulations.** Specific HC formulations are listed on the y-axis (see Box 1 for definitions), while the HC formulation usage period length in days (days shown on the logarithmic scale) is plotted along the x-axis. The colour illustrates the HC type. The box represents the interquartile range (25th to 75th percentile), with the central diamond shapes marking the median, and the black dots marking outliers. Abbreviations: IUD – intrauterine device; COC, combined oral contraceptive; CHC, combined hormonal contraceptive; (s), CHC pill with sequential preparations; (p), transdermal patch.

### Hormonal contraceptive switching

While 35.8% ($n_{users}$ = 26,151) of women used only one HC formulation during the observed study period, 64.2% ($n_{users}$ = 46,920) switched to a different one (ranging between 2 and 11 different HC formulations, Fig 4A). Specifically, out of all HC users, 23.9% ($n_{users}$ = 17,453) were classified as non-switchers, 46.5% ($n_{users}$ = 33,991) as broad switchers, 17.7% ($n_{users}$ = 12,929) as rapid switchers, 7.3% ($n_{users}$ = 5,362) as rapid discontinuers, while 4.6% ($n_{users}$ = 3,336) did not fit into any of the defined categories. The latter group included, for example, users of one HC formulation for less than 180 days, but more than 90 days (see Methods). From these types of users, we were interested in non-switchers, rapid switchers and rapid discontinuers (user characteristics presented in Table E in S3 Appendix). Non-switchers and rapid switchers were predominantly women from the 20–29 age group (33.8% and 48.8%, respectively), while rapid discontinuers were from

PLOS Medicine

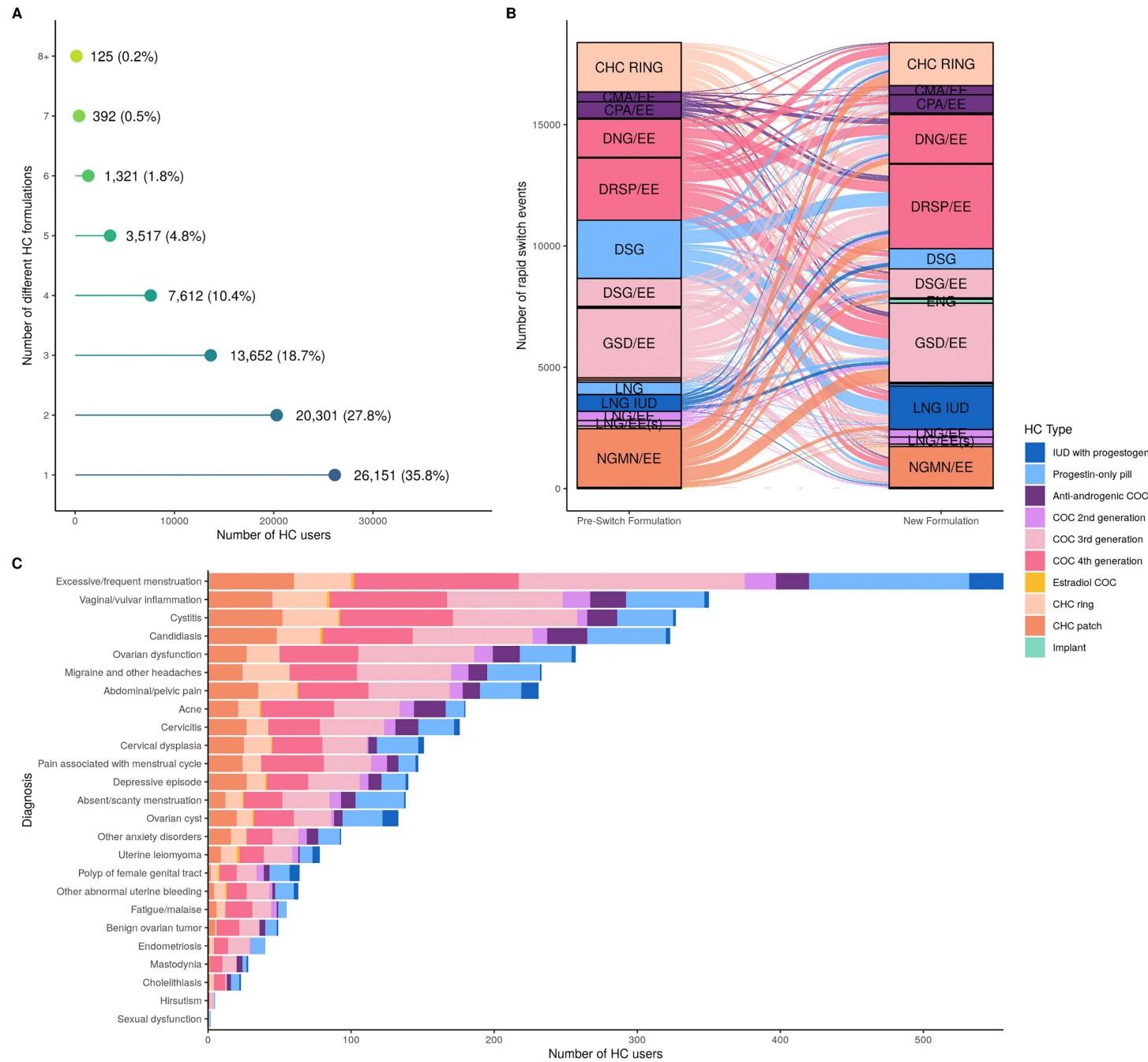

**Fig 4. Hormonal contraceptive (HC) usage behaviour. A**: Number of different HCs purchased per individual during the study observation period. **B**: Graphical illustration of switching pathways of rapid switchers (pre-switch formulation is the HC method individuals switched from, while the new formulation is the HC method individuals switched to; see Box 1 for HC formulation abbreviations). The colour illustrates the HC type. **C**: Main ICD-10 diagnoses recorded between the first purchase of pre-switch HC formulation and the new formulation in rapid switchers. The coloured bars indicate the type of pre-switch HC method. Abbreviations: IUD, intrauterine device; COC, combined oral contraceptive; CHC, combined hormonal contraceptive.

the 30–39 age group (32.2%). The non-switcher usage periods ranged from 182 to 6,934 days, with an average of 1,658 days (corresponding to 4.5 years), whereas in rapid switchers, the average time to switch from one formulation to another was 49 days.

The illustration of the HC switching behaviour of rapid switchers with complex multidirectional flows is presented in Fig 4B. The HC formulations with the highest number of rapid switch events were short-acting HCs with the highest usage volume, such as GSD/EE, DRSP/EE, norelgestromin and ethinylestradiol (NGMN/EE) patch, DSG, and CHC ring. Each of these accounted for 14.6%–20.4% of rapid switchers who had at least one rapid switch from that formulation, and around 9% of users of given HC, except 17.3% of NGMN/EE users, and around 14% of DSG, and CHC ring users. The subsequent HC formulations for these switchers were similar and overlapping, where GSD/EE rapid switchers mostly switched to DRSP/EE (25.2%), dienogest/ethinylestradiol (DNG/EE, 14.7%), and NGMN/EE patch (11.0%), DRSP/EE users switched to GSD/EE (23.3%), DNG/EE (16.4%), and CHC ring (13.2%), and NGMN/EE users switched to GSD/EE (23.7%), CHC ring (20.6%) and DRSP/EE (19.4%). Most notably, switching to DNG/EE, LNG-IUD, and ENG implant were more common than switching from them. Among rapid switchers who switched from COCs ($n_{users}$ = 7,800 (60.3%)), 66.5% switched to a different COC, 11.3% switched to a CHC patch, and 11.1% to the CHC ring. Those who switched to the ring used the new method for approximately 4.9 months (median value) compared with 3.7 months for a patch.

In the period between the first purchase date of the pre-switch HC formulation and the first purchase date of the new formulation, no ICD-10 diagnosis codes were identified for 50.2% ($n_{users}$ = 6,486) of rapid switchers, while 23.1% ($n_{users}$ = 2,982) had diagnoses from our curated list of potential side effects (see Methods). Notably, we observed diagnoses related to reproductive system disorders (e.g., excessive/frequent menstruation ($n_{users}$ = 556)), neurological/pain conditions (migraine and other headaches ($n_{users}$ = 233), acne ($n_{users}$ = 180), menstrual pain ($n_{users}$ = 147)), mood disorders (depressive episode ($n_{users}$ = 140), anxiety ($n_{users}$ = 93)), and mastodynia ($n_{users}$ = 28) (Fig 4C and Table F in S3 Appendix). In addition to this, HC switching was previously associated with contraceptive failure and subsequent abortion [63]. In our cohort, medical abortion (ICD-10 O04) overlapped with inferred HC usage periods in 3.2% ($n_{users}$ = 2,359, $n_{periods}$ = 2,440) of all users, with COCs being the most used contraceptive before abortion. The majority of these women (77.3% ($n_{users}$ = 1,823)) used HCs again (median time to the new HC formulation = 684 days/1.9 years (11–6,132 days)), where other COCs ($n_{periods}$ = 792 (32.5%)), LNG-IUD ($n_{periods}$ = 125 (5.1%)) and POPs ($n_{periods}$ = 122 (5.0%)) were the most common choices following abortion. Similarly, in 2.2% ($n_{rapid}$ = 288, $n_{periods}$ = 291) of rapid switchers, abortion diagnosis was recorded between purchases of two HC formulations. The most common switches were from one short-acting method to another of comparable effectiveness and safety, such as from one COC to a different COC ($n_{periods}$ = 79 (27.1%)), from a CHC patch to a COC ($n_{periods}$ = 43 (14.7%)), and from a COC to the CHC ring ($n_{periods}$ = 28 (9.6%)).

## Presence of thromboembolism risk factors

Among 13,610 HC users with purchases in 2022, which is the most recent year with complete data, 65.0% ($n_{users}$ = 8,840) were CHC users (mean age at purchase (SD) = 33.9 (8.1) years; median = 33, IQR [27,40]) and 35.0% ($n_{users}$ = 4,770) POC users (mean age at purchase = 37.0 (7.8) years; median = 37, IQR [31,43]), with 590 individuals using both types. Individuals who used different HCs in 2022 were classified based on their first HC. Thromboembolism was diagnosed in 43 women (VTE/ATE diagnosis recorded after the first HC purchase date in 2022; mean age at diagnosis = 40.0 (7.5) years; median = 40, IQR [35,45]), of whom 38 were diagnosed during the inferred HC usage period. Next, to characterise the thromboembolism risk profile of POC and CHC user groups, we calculated the prevalence ratio (PR) for 15 different risk factors (Fig 5 and Table G in S3 Appendix, with POC users as the reference category). The most common risk factors were age above 35 (55.1% of POC users and 40.6% of CHC users; PR = 0.74, 95% CI [0.71, 0.77]; $p_{Bonferroni}$ < 0.001), purchase of risk-increasing medications (14.8% of POC and 17.2% of CHC; PR = 1.16, 95% CI [1.07, 1.26]; $p_{Bonferroni}$ = 0.006), obesity (14.5% of POC and 10.0% of CHC; PR = 0.69, 95% CI [0.63, 0.76]; $p_{Bonferroni}$ < 0.001), and migraine (8.2% of POC and 5.6% of CHC; PR = 0.68, 95% CI [0.60, 0.78]; $p_{Bonferroni}$ < 0.001). The prevalence for carrier status of thrombosis

PLOS Medicine

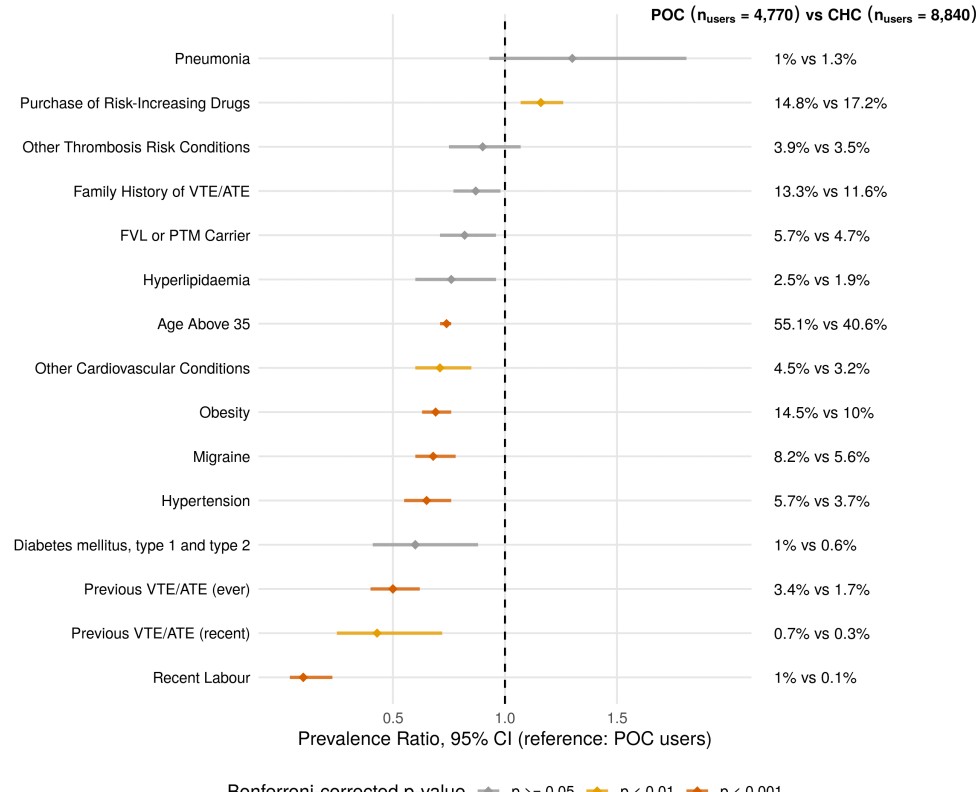

**Fig 5. Prevalence ratios of venous and arterial thromboembolism (VTE and ATE) risk factors in progestin-only hormonal contraceptive (POC) vs. combined hormonal contraceptive (CHC) users with purchases in 2022.** On the y-axis are risk factors and within-group percentages, while on the x-axis is the prevalence ratio (PR) with 95% confidence interval (CI). Abbreviations: FVL, Factor V Leiden; PTM, Prothrombin G20210A Mutation.

risk-increasing genetic variants was slightly higher among POC users (5.7% of POC versus 4.7% of CHC; PR = 0.82, 95% CI [0.71, 0.96]; $p_{Bonferroni}$ = 0.167), while the prevalence of personal history of VTE/ATE was twice as high (3.4% of POC and 1.7% of CHC; PR = 0.50, 95% CI [0.40, 0.62]; $p_{Bonferroni}$ < 0.001). The prevalence of family history of thrombosis, other thrombosis risk conditions, hyperlipidaemia, diabetes, and pneumonia did not differ significantly between POC and CHC users after Bonferroni correction ($p \geq 0.003$). The same analysis, stratified by estrogen dose (low ethinylestradiol, moderate ethinylestradiol, and estradiol CHC user groups), showed largely similar patterns across low and moderate ethinylestradiol groups (Table H in S3 Appendix and S6 Appendix). The estradiol group showed slightly higher prevalence ratios across several risk factors, though estimates should be interpreted cautiously given the small sample size and wide confidence intervals. Out of 13,610 analysed HC users, 61.7% ($n_{users}$ = 8,397) had at least one first-degree relative in the EstBB, and 7.5% ($n_{users}$ = 1,022) had a first-degree relative diagnosed with VTE or ATE before HC purchase. Women using HCs who had at least one first-degree relative diagnosed with thromboembolism had 1.7-fold higher odds of being a FVL or PTM carrier (76/1,022 (7.4%) versus 327/7,375 (4.4%); OR = 1.73, 95% CI [1.32, 2.25]; $p < 0.001$). The sensitivity and specificity of family history of thrombosis for identifying carrier status were 18.9% (95% CI [15.0, 22.7]) and 88.2% (95% CI [87.5, 88.9]), respectively.

Across the entire study group of all HC users ($n_{users}$ = 73,071), we found that 5.3% ($n_{users}$ = 3,886) carried at least one VTE high-risk genetic variant (3.5% ($n_{users}$ = 2,556) only FVL, 1.8% ($n_{users}$ = 1,276) only PTM, 0.07% ($n_{users}$ = 54) both FVL and PTM). While in 1.5% ($n_{users}$ = 1,060) of users, thrombosis diagnosis occurred during the inferred HC usage period, of

those, only 17.6% purchased antithrombotic agents (ATC group B01A) after the diagnosis and while HCs were in active use according to purchasing data. The top five ICD-10 thrombosis codes were I80 (unspecified phlebitis and thrombophlebitis, $n_{users}$ = 330), I80.2 (deep vein thrombosis, $n_{users}$ = 212), I26.9 (pulmonary embolism, $n_{users}$ = 130), I80.3 (embolism or thrombosis of lower extremity, $n_{users}$ = 125) and I82.8 (embolism and thrombosis of vena cava, $n_{users}$ = 78). Among the 1,060 cases during HC usage, 8.6% carried either the FVL or PTM variant. Overall, we observed a significantly higher percentage of thrombosis cases at any time point in carriers using HC (6.5%) compared to the non-carriers using HC (4.2%; OR = 1.61, 95% CI [1.40, 1.84], $p < 0.001$).

## Discussion

In this study, using EstBB data and prescribed HC purchasing data from the period 2004–2022, we aimed to describe in detail HC use trends of >73,000 EstBB female participants, compare results with the Estonian population HC use trends and evaluate the suitability of longitudinal biobanks, such as EstBB, for studying genetic risk for HC side effects. Our described use trends of various HCs were generally consistent with the population-level statistics, albeit with slight differences discussed below. We identified several HC user types, such as non-switchers, broad switchers, rapid switchers, and rapid discontinuers. While 64.2% of women using HC eventually switched to a different formulation, 17.7% switched within 3 months of their first purchase, which was previously associated with side effect-driven switching [6,7]. Indeed, we detected first-time records of ICD-10 diagnosis codes related to potential HC side effects in 23.1% of rapid switchers, with excessive/irregular menstrual bleeding being the most common. We also estimated the prevalence of genetic variants associated with higher VTE risk and evaluated various VTE and ATE risk factors, including personal and family history of thrombosis.

Compared with Estonian national statistics [13], we had slightly higher user prevalence for POCs and CHCs (Fig 2B). This could be due to the several methodological differences, such as differences in age range (15–49 versus 15–55 years), denominator (Estonian population versus EstBB female participants), and the definition of HC use, where we set maximum length for long-acting methods, included cyproterone/ethinylestradiol (CPA/EE) HC, and incorporated short pauses (<90 days between two purchases) as part of the usage period. Also, the EstBB 15–29 age groups have drastically decreased in size in recent years (Fig 2A); therefore, the user prevalence became "artificially" higher, as no new participants were recruited into EstBB after 2019. Despite these differences, the similar temporal trends suggest that the overall pattern of HC usage is consistent (Fig 2B). Next, regarding higher implant use among younger women, increased POC use in recent years, the presence of the most common risk factors and POC/CHC use among women with risk factors, our results are in line with those from Kurvits and colleagues [13]. Notably, thromboembolism risk factors were more prevalent among POC users than CHC users (Fig 5), reflecting confounding by indication, as individuals with multiple risk factors are preferentially prescribed POCs. However, the concomitant purchase of risk-increasing medications was statistically more prevalent among CHC users. This may highlight areas for improvement. Finally, we observed that some women continued to purchase HCs even after the thrombosis diagnosis, without the concomitant purchase of antithrombotic agents [64]. However, this finding should be interpreted cautiously, since medication purchase data does not reflect medications administered during inpatient care.

Several observational studies evaluating contraceptive use in Nordic countries [65,66] and the UK [67] reported increasing use of levonorgestrel-containing COCs, POPs and IUDs. However, in EstBB, we observed an increase only in IUD use (Fig 2C and Table D in S3 Appendix), which aligns with observations in the Estonian population [13]. We note that the annual user prevalence may reflect regulatory changes, prescription practices, and market availability, as not all HC formulations were available throughout the whole study observation period. Specifically, over the 19-year study period, HC options in Estonia evolved in line with the EMA guidelines [10] and under regulatory oversight of the State Agency of Medicines (Ravimiamet) [68]. Key changes included a shift towards low-dose estrogen COCs, and improving access to long-acting methods and emergency contraceptives. For example, subdermal implants got marketing authorisation in

Estonia in December 2013, thereby diversifying the available contraceptive methods from 2014. Furthermore, shifts in clinicians' thinking over the study period, such as broader use of IUDs in nulliparous women and POPs beyond lactating women, likely contributed to the observed trends. Next, a study investigating prescription rates of contraceptives during the COVID-19 pandemic in England discovered an overall significant decrease in prescriptions, and an increase in POPs prescriptions [62], which might be relevant to consider when analysing use trends during this period. While we focussed on purchasing and not prescription dates, we observed a higher number of CHC and POP purchases right before lockdown, likely due to stockpiling behaviour, and a lower number during the first year of the COVID-19 pandemic, especially of implants and LNG-IUDs, likely due to limited access to healthcare services. However, these differences were not notable (S5 Appendix). Estonia has a well-organised and digitalised healthcare system, and there has never been a complete lockdown. Also, the ageing cohort could partially explain the gradual decrease in HC initiators; while the number of women using HCs globally is growing as the population rises and renews [1], the number of women at peak reproductive age in the EstBB is decreasing.

While population-based studies are great for describing use trends and general risk factors, they often lack genetic data to evaluate the presence of specific large-effect genetic risk factors. Here, population-based biobanks with readily available genetic data have an advantage. It has been shown that up to 20% of individuals diagnosed with VTE (25% in the case of idiopathic VTE) carry at least one genetic variant associated with increased risk [20]. We found that 5.3% of women using HC carried either FVL or PTM variant. At the same time, generally, only family history is evaluated before HC prescription, serving traditionally as a proxy for genetic risk. However, family history of thrombosis does not always adequately reflect carrier status [69,70]. Readily available information on carrier status in biobanks can potentially reduce the incidence of VTE among women on HC. In parallel, stratification based on VTE and ATE polygenic risk scores could possibly offer a more comprehensive approach to identifying women who require more careful contraceptive prescribing [23] and are eligible for additional thrombophilia screening or normalised activated protein C sensitivity ratio (nAPCsr) blood coagulation testing.

When it comes to large-scale genetic analyses of drug use and associated side effects, defining cases and controls is the first step, but, unlike for cardiometabolic medications with clear physiological endpoints, doing so for hormonal medications can be challenging. Using a similar approach from Lo and colleagues [71], but contextually adapted, we identified several HC user types, such as non-switchers, broad switchers, rapid switchers, and rapid discontinuers that could be used as phenotypes of interest. In general, switching between different HC formulations is a common part of women's contraceptive journey throughout their reproductive years [44]. However, rapid discontinuation or switching to a different HC formulation can be due to the side effects [6,7]. For that reason, we focussed on rapid switchers and investigated ICD-10 diagnoses recorded between the first purchase of pre-switch HC formulation and the first purchase of post-switch HC formulation. Indeed, we found records of diagnoses related to potential side effects. The observed prevalence of these diagnoses among only 23.1% of rapid switchers could potentially be explained by underreporting or other reasons for switching, and a different study design is needed to clarify this. Similarly, from inferred usage periods of HC initiators, we observed the highest number of women on short-acting methods clustered around 90 days period length (Fig 3). This aligns with the general recommendation to use HCs for at least three months to allow the body to adapt to exogenous hormones [72,73], but it also suggests that, for many women, initial HC method matching is suboptimal. This highlights the need for developing more effective approaches to HC prescribing, potentially accounting for individual genetic risk as a modifier of side effects associated with HC use.

The key strength of our study is the high-quality EstBB datasets derived from multiple sources, including genetic and detailed clinical information, that enable comprehensive analysis of individual HC use over a period of more than 10 years. Additionally, our data and approach for defining HC exposure using purchase records avoid self-report and recall bias in HC usage duration and timing, which have been shown to increase errors and reduce research reproducibility significantly [74,75].

Our study has several limitations that need to be mentioned, as it relies on medication purchase data that were not primarily collected for research. First, the described HC trends are specific to Estonia, and the investigated individuals are of European ancestry, which may not apply to other healthcare systems and ancestries. Second, the inferred HC usage periods derived from purchase data may not accurately reflect actual usage or adherence to the HC regime. This may lead to a slight overestimation of the usage periods. However, the short-acting HC effects extend beyond inconsistent adherence, as body gradually adapts to the hypothalamic-pituitary-gonadal axis's natural rhythm after discontinuation. Third, we could not reliably track the removal of the long-acting reversible contraceptives with the ICD-10 coding system; we were able to censor only by using pregnancy dates. The distinction of separate four-character subcategory codes for IUD and implant insertion, checking, and removal by the medical community would benefit future observational studies. Fourth, we did not have access to the information on purchases of emergency contraceptives and injections. Emergency contraceptives are sold 'over-the-counter' in Estonia, and our input was purchases of prescribed HCs. The injectable medroxyprogesterone is officially an unauthorised product and is used only on the specific doctor's recommendation; therefore, we anticipated a negligible number of users. Fifth, given the current research setting, which differs from the gold-standard Phase 3 clinical studies, we could not definitively determine the indication for use, nor the specific reasons for switching between formulations or discontinuation. Future implementations of free-text data mining of unstructured EHRs could partially aid in mitigating this limitation. Additionally, though out-of-pocket expenses could not be captured and may have impacted usage trends and switching, this is unlikely to be substantial, given that EHIF reimburses at least 50% of the full price for HCs [76]. Sixth, we acknowledge that the presented numbers are study population averages, while HC research requires more personalised insights [77]. However, our valuable cohort-level statistic, compared to the national-level statistic, provides a baseline that enables meaningful interpretation of individual variation. At the same time, population-based biobanks facilitate future personalised approaches to HC prescribing.

In conclusion, we reconstructed individual HC usage periods from HC purchase data across a 19-year study period and linked them to users' genetic and health data. Our analyses reveal real-world longitudinal HC use, with nuances across formulations, including frequent formulation switching and discontinuation. By demonstrating that EstBB trends align with national statistics while capturing detailed individual HC use trajectories, user profiles, and potential side effect-related diagnoses, this study supports the use of population-based biobanks as a valuable resource for HC research. Expanding on our approach, future studies could analyse side effects stratified by the number of switches, and investigate the impact of genetic factors on switching behaviour and HC-related side effects, ultimately enabling more personalised HC prescribing.

## Supporting information

**S1 Checklist. STROBE and RECORD checklists. Table A.** STROBE checklist. This checklist is reproduced from the Strengthening the Reporting of Observational Studies in Epidemiology (STROBE) Statement, which is licensed under the Creative Commons Attribution 4.0 International (CC BY 4.0). Information about the STROBE Initiative is available at https://www.strobe-statement.org/. von Elm E, Altman DG, Egger M, Pocock SJ, Gøtzsche PC, Vandenbroucke JP, et al. (2007) The Strengthening the Reporting of Observational Studies in Epidemiology (STROBE) Statement: Guidelines for Reporting Observational Studies. PLOS Medicine 4(10): e297. https://doi.org/10.1371/journal.pmed.0040296. **Table B**. RECORD checklist. This checklist is reproduced from the REporting of studies Conducted using Observational Routinely-collected Data (RECORD) Statement, which is licensed under the Creative Commons Attribution 4.0 International (CC BY 4.0). Information about the RECORD Initiative is available at https://www.record-statement.org/. Benchimol EI, Smeeth L, Guttmann A, Harron K, Moher D, Petersen I, et al. (2015) The REporting of studies Conducted using Observational Routinely-collected health Data (RECORD) statement. PLOS Medicine 12(10):e1001885. https://doi.org/10.1371/journal.pmed.1001885.
(XLSX)

**S1 Appendix. Illustration of the inferred HC usage periods of three distinctive hormonal contraceptive (HC) user types of interest: non-switchers, rapid discontinuers and rapid switchers.** The manually defined data points are for illustrative purposes and are not real observations.
(TIF)

**S2 Appendix. Illustration of the strategy for identifying relevant International Classification of Diseases, 10th Revision (ICD-10) diagnosis codes and thromboembolism risk factors. Fig A:** identification of ICD-10 diagnosis codes (please see Table A in S3 Appendix for diagnosis names). **Fig B:** identification of thromboembolism risk factors (please see Table A in S3 Appendix for diagnosis names).
(TIF)

**S3 Appendix. Supporting information. Table A.** ICD-10 codes used to identify HC initiation/discontinuation reasons, and thromboembolism events and risk factors. **Table B.** Total number of individuals from specific age group in the whole study cohort. The table includes both hormonal contraceptive (HC) users and HC non-users. **Table C.** Annual prevalence of progestin-only hormonal contraceptive and combined hormonal contraceptive user groups in the Estonian Biobank (EstBB, for years from 2004 to 2022) and the Estonian population (Kurvits and colleagues [13], for years from 2005 to 2019). **Table D.** Total covered days by hormonal contraceptive type and age group per calendar year (from 2004 to 2022). COC, combined oral contraceptive; CHC, combined hormonal contraceptive; IUD, intrauterine device. **Table E.** Characteristics of hormonal contraceptive user types of interest (non-switchers ($n = 17{,}453$), rapid switchers ($n = 12{,}929$) and rapid discontinuers ($n = 5{,}362$)). The numbers refer to the earliest observed relevant HC usage period. COC, combined oral contraceptive; CHC, combined hormonal contraceptive; IUD, intrauterine device. **Table F.** Count of rapid switchers with potential side effect-related diagnoses from a curated ICD-10 code list, by hormonal contraceptive type. COC, combined oral contraceptive; CHC, combined hormonal contraceptive; IUD, intrauterine device. **Table G.** Prevalence of thromboembolism risk factors (RF) and prevalence ratio (PR) with 95% confidence interval (CI) between combined hormonal contraceptive (CHC) and progestin-only contraceptive (POC) users, with raw and Bonferroni-corrected $p$-values. **Table H.** Prevalence of thromboembolism risk factors (RF) and prevalence ratio (PR) with 95% confidence interval (CI) between combined hormonal contraceptive (CHC) users stratified by estrogen dose (low ethinylestradiol (low_ee), moderate ethinylestradiol (moderate_ee), and estradiol (e2)) and progestin-only contraceptive (POC) users, with raw and Bonferroni-corrected $p$-values.
(XLSX)

**S4 Appendix. Percentage of hormonal contraceptive (HC) types among initiators across studied years, stratified by age groups.** Abbreviations: IUD, intrauterine device; COC, combined oral contraceptive; CHC, combined hormonal contraceptive.
(TIF)

**S5 Appendix. Hormonal contraceptive (HC) purchase trends in 2019 (pre-Coronavirus Disease 2019 (COVID-19) period), 2020 and 2021 (COVID-19 periods). Fig A:** Monthly deviation ratio from the yearly average for years 2005–2021 (exact value marked with the dot). The red colour and line mark 2020, while the black and grey lines mark the mean across all years and ±3 standard deviations, respectively. **Fig B:** Comparison of the total number of HC purchases per month in the years 2019 (light blue line), 2020 (red line) and 2021 (dark blue line). **Fig C:** The total number of monthly HC purchases in 2019, 2020 and 2021, stratified by HC type. Abbreviations: IUD, intrauterine device; COC, combined oral contraceptive; CHC, combined hormonal contraceptive. **Fig D:** The total daily number of HC purchases in March 2019, 2020, and 2021. The number 13 is coloured red to mark the date when the Estonian government announced a state of emergency in 2020.
(TIF)

**S6 Appendix. Prevalence ratios of venous and arterial thromboembolism (VTE and ATE) risk factors in progestin-only hormonal contraceptive (POC) versus combined hormonal contraceptive (CHC) users with purchases in 2022.** CHC users were also stratified by estrogen dose: low-dose ethinylestradiol (low-EE), moderate-dose EE (moderate-EE), and estradiol (E2). On the y-axis are risk factors, while on the x-axis is the prevalence ratio (PR) with 95% confidence interval (CI). Abbreviations: FVL, Factor V Leiden; PTM, Prothrombin G20210A Mutation.
(TIF)

## Acknowledgments

The authors thank the Estonian Biobank participants for their contribution to this work, and the Estonian Biobank Research Team members (Andres Metspalu, Tõnu Esko, Mait Metspalu, Mari Nelis and Georgi Hudjashov) for data collection, genotyping, QC and imputation. We also thank the High-Performance Computing Center of the University of Tartu for providing computational resources for data analysis, and the Health Informatics research team (Raivo Kolde, Sven Laur, Sulev Reisberg and Jaak Vilo) for data harmonisation, mapping to OMOP CDM, and fact extraction.

## Author contributions

**Conceptualisation:** Jelisaveta Džigurski, Hannah Currant, Reedik Mägi, Lili Milani, Triin Laisk.

**Data curation:** Jelisaveta Džigurski.

**Formal analysis:** Jelisaveta Džigurski.

**Funding acquisition:** Triin Laisk.

**Investigation:** Jelisaveta Džigurski, Hannah Currant, Mall Eltermaa, Reedik Mägi.

**Methodology:** Jelisaveta Džigurski, Märt Möls, Kristi Läll.

**Resources:** Reedik Mägi, Lili Milani.

**Supervision:** Märt Möls, Kristi Läll, Reedik Mägi, Lili Milani, Triin Laisk.

**Visualisation:** Jelisaveta Džigurski.

**Writing – original draft:** Jelisaveta Džigurski.

**Writing – review & editing:** Jelisaveta Džigurski, Märt Möls, Kristi Läll, Hannah Currant, Mall Eltermaa, Reedik Mägi, Lili Milani, Triin Laisk.

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
