## [Editor Report · Decision Letter 0]

9 Dec 2025

Dear Dr Laisk,

Thank you for submitting your manuscript entitled "Hormonal contraceptive use trends in the Estonian Biobank: user profiles and a resource for genetic studies" for consideration by PLOS Medicine.

Your manuscript has now been evaluated by the PLOS Medicine editorial staff, and I am writing to let you know that we would like to send your submission out for external peer review.

For clinical studies, please upload a copy of your trial study protocol as a supporting information file. The study protocol should be the version submitted for approval to the institutional review board or ethics committee, should include any amendments to the study protocol, as well as the date of their approval by the institutional review or ethics committee. Please also detail any deviations from the study protocol in the Methods section of your manuscript. The editors will consider the protocol and study conduct prior to a final decision for external review.

Please re-submit your manuscript within two working days, i.e. by Dec 11 2025 11:59PM.

Kind regards,

Helen Howard

PLOS Medicine

---

## [Decision Letter · Decision Letter 1]

29 Jan 2026

Dear Dr Laisk,

Many thanks for submitting your manuscript "Hormonal contraceptive use trends in the Estonian Biobank: user profiles and a resource for genetic studies" (PMEDICINE-D-25-04320R1) to PLOS Medicine. The paper has been reviewed by subject experts and a statistician; their comments are included below and can also be accessed here: [LINK]

As you will see, the reviewers found this work to be a valuable addition to the existing knowledge and the manuscript to be well-written. Clarifications and further explanations were required to improve the clarity of the manuscript. A number of methodological concerns that need attention were raised as well, especially in regards to HC medication switches, confounding by indication and the non-independency of women in the cohort. You can find the detailed report of reviewers' comments at the end of this letter. After discussing the paper with the editorial team and an academic editor with relevant expertise, I'm pleased to invite you to revise the paper in response to the reviewers' comments. We plan to send the revised paper to some or all of the original reviewers, and we cannot provide any guarantees at this stage regarding publication.

We ask that you submit your revision by Feb 16 2026 11:59PM. However, if this deadline is not feasible, please contact me by email, and we can discuss a suitable alternative.

Don't hesitate to contact me directly with any questions (efourli@plos.org).

Best regards,

Evangelia

Evangelia Fourli, Ph.D.

Associate Editor

PLOS Medicine

efourli@plos.org

Comments from the editorial team:

- Please comment on potential selection bias by including only people covered by national health insurance.

- Moreover, we ask that you discuss as a limitation the lack of information regarding the underlying reasons for switching HCs, e.g. are there differences in cost even under insurance?

-Finally, can you please discuss if, in this 20 year period assesses, there were major changes in HC options that may drive such HC changes?

Comments from the reviewers:

Reviewer #1: Thank you for the opportunity to review the manuscript titled, "Hormonal contraceptive use trends in the Estonian Biobank: user profiles and a resource for genetic studies". My specific comments are below --

- Page 2, lines 22-23: This may be a difference between US prescriptions and Estonian, but can the authors define "purchase data". Does this mean prescribing data? More detail would be helpful. Purchase data to me would mean over-the-counter or medications that can be purchases without a prescription from a physician.

- Page 2, line 30-31: Please provide the descriptive statistics for this sentence. Do you mean combined HCs dominated the total prescriptions? More information here would be helpful.

- Page 2, line 32: Provide the number of "users" for these percentages. It may also be good to differentiate between "users" and "participants" or use one term consistently.

- Page 4, line 78-80: Please cite the WHO guidelines in this sentence.

- Page 4, line 85: It would be helpful to provide the total number of participants in the Biobank - I assume this is not exclusively women?

- Page 4, line 95: In all biobanks or specifically the EstBB?

- Page 5, line 110: Provide the current total population of Estonia.

- Page 6, line 127L Just to confirm, the purchase invoice is the individual picking up the prescription at a pharmacy? Please clarify.

- Page 8, lines 168-175: Where these categories created by the author team or an already defined set of categories from other research?

- Page 13: Thank you for the beautiful figures - I love them!

- Page 17, lines 322-341: This section is a mix of result findings and discussion. Please consider moving some of the interpretation of the data to the discussion.

- Page 26, line 503: Self-report would be a disadvantage when it comes to specific symptom side effects of HC exposure.

Reviewer #2: The manuscript presents an interesting and comprehensive analysis of hormonal contraceptive (HC) use among participants from the Estonian Biobank. The study is well-written and represents a significant contribution to understanding real-world patterns of HC use. Notable strengths include the longitudinal approach, with an 18-year observation period, and the integration of application data with biobank records. In addition, the manuscript has a high relevance for clinical practice and public health.

Critical Comments and Suggestions for Improvement:

1. General Use Trend:

The authors note that not all HC formulations were available throughout the entire observation period (line 280ff.). It would be beneficial to clarify whether, and to what extent, this fact influences the annual prevalence rates.

2. Contraceptive Switching:

The distinction between "rapid switchers" and "rapid discontinuers" is clear. However, Figure 3a shows that individuals switched between one and more than eight formulations. It would strengthen the manuscript if the authors analyzed potential side effects stratified by the number of switches.

3. Terminology:

The term "broad switcher" is used in the results and discussion sections but is not defined in the methods. Please clarify whether this term is synonymous with "switcher" or if it has a distinct meaning.

4. Visualization of Switching Pathways:

Figure 3b, which illustrates switching pathways, is difficult to interpret due to the numerous changes and color coding. I recommend including the detailed information on HC switching behavior from Additional File 1 in the main text.

5. Side Effects and ICD-10 Diagnoses:

The manuscript should clarify whether the listed ICD-10 diagnoses appeared in the patient's medical records for the first time before the change in medication, or if they had already occurred before the first contraceptive was taken. Additionally, please specify what period prior to contraceptive use was considered.

Conclusion:

Overall, the manuscript is well-executed and provides valuable insights. Addressing the points above would further enhance the clarity and impact of the findings.

Reviewer #3: This manuscript addresses an important public health topic and is based on a new and valuable nationwide dataset with substantial potential to inform contraceptive use patterns and safety. The manuscript is well written; therefore, my comments are intended to strengthen clarity of the manuscript.

-The manuscript states that purchases with dosages <0.3 or >6 were excluded to avoid mistype errors. While the rationale is generally reasonable, the explanation should be more explicit linked to the Estonian dispensing and prescribing system. Specifically, it would be helpful to write if this rule applies uniformly across all HC products or depends on prescription duration, or clinical context.

-Although only 0.04% of purchases were excluded, it would be helpful to explicitly state that this filtering step had a negligible impact on prevalence estimates and downstream analyses, as even small exclusions can raise concerns about selection bias in population-based studies.

-The adjustment of HC exposure periods using pregnancy-related ICD-10 codes require further information. For example, the authors should explain if HC exposure periods were truncated at the first pregnancy-related code, fully excluded, or handled in another way, and how multiple or overlapping pregnancy codes were treated.

-The statement that "usage periods were created using Node.js v18.12.1" is unclear and does not add meaningful methodological information. Therefore, the authors should explain what Node.js was used (e.g., execution of custom data-processing scripts, prescription linkage, exposure window construction), and if a bespoke script or algorithm was developed. If such a script exists, indicating if it is available (e.g., in supplementary materials or a repository), or explaining why it cannot be shared.

- The manuscript reports that BMI ranged from 14.1 to 43 with a mean of 24.3 during HC use. However, this range appears to reflect the predefined inclusion criteria rather than observed variability in the data. The wording should therefore indicate that this was the eligible BMI range.

- The authors should describe briefly in the text how BMI measurements were obtained in relation to HC use (e.g., closest measurement before first initiation or averaged over time) and if it was self-reported.

- The authors observed that estimated HC prevalence rates were systematically higher than those reported by Kurvits, although the temporal trends were similar. To aid interpretation, the authors should explain briefly the key differences between the two studies in the text, such as age ranges, data sources, denominators, and definitions of HC use.

- The authors reported a higher prevalence of thromboembolism risk factors among POC users. This finding reflects confounding by indication, as individuals with elevated thrombosis risk are preferentially prescribed progestogen-only methods. Without appropriate adjustment, comparisons between POC and CHC users are inherently biased. I suggest that the authors conduct additional analyses adjusting for age, BMI, smoking status, comorbidities and prior thrombosis when comparing the risk profiles of the HC groups.

- The study would benefit from being stratified by estrogen dose. Given the well-established dose-dependent relationship between estrogen and thrombosis risk, stratifying by estrogen dose (e.g. ≤20 µg versus >20 µg ethinylestradiol, or other ranges) would enhance its clinical relevance.

Reviewer #4: See attachment

Michael Dewey

---

* Please upload any figures associated with your paper as individual TIF or EPS files with 300dpi resolution at resubmission; please read our figure guidelines for more information on our requirements: http://journals.plos.org/plosmedicine/s/figures. While revising your submission, we strongly recommend that you use PLOS's NAAS tool (https://ngplosjournals.pagemajik.ai/artanalysis) to test your figure files. NAAS can convert your figure files to the TIFF file type and meet basic requirements (such as print size, resolution), or provide you with a report on issues that do not meet our requirements and that NAAS cannot fix.

After uploading your figures to PLOS's NAAS tool - https://ngplosjournals.pagemajik.ai/artanalysis, NAAS will process the files provided and display the results in the "Uploaded Files" section of the page as the processing is complete.

If the uploaded figures meet our requirements (or NAAS is able to fix the files to meet our requirements), the figure will be marked as "fixed" above. If NAAS is unable to fix the files, a red "failed" label will appear above.

When NAAS has confirmed that the figure files meet our requirements, please download the file via the download option, and include these NAAS processed figure files when submitting your revised manuscript.

* Please ensure that the study is reported according to the STROBE guideline and include the completed STROBE checklist as Supporting Information. When completing the checklist, please use section and paragraph numbers, rather than page numbers. Please add the following statement, or similar, to the Methods: "This study is reported as per STROBE guideline (S1 Checklist)."

FIGURES AND TABLES

SUPPLEMENTARY MATERIAL

REFERENCES

OBSERVATIONAL STUDIES

* Abstract: Please include the study design, population and setting, number of participants, years during which the study took place (enrollment and follow up), length of follow up, and main outcome measures.

* Please ensure that the study is reported according to the STROBE (or appropriate STOBE extension) guideline (available from: https://www.equator-network.org/reporting-guidelines/strobe) and include the completed STROBE (or STROBE extension) checklist as Supporting Information. Please add the following statement, or similar, to the Methods: "This study is reported as per the Strengthening the Reporting of Observational Studies in Epidemiology (STROBE) guideline (S1 Checklist)." When completing the checklist, please use section and paragraph numbers, rather than page numbers.

* [FOR POPULATION HEALTH/REGISTRY STUDIES] Please ensure that the study is reported according to the RECORD guideline (available from https://www.record-statement.org) and include the completed checklist as Supporting Information. Please add the following statement, or similar, to the Methods: "This study is reported as per the Reporting of Studies Conducted using Observational Routinely-Collected Data (RECORD) guideline (S1 Checklist)." When completing the checklist, please use section and paragraph numbers, rather than page numbers.

* [FOR POPULATION HEALTH ESTIMATES] Please ensure that the study is reported according to the GATHER statement (available from https://www.equator-network.org/reporting-guidelines/gather-statement) and include the completed checklist as Supporting Information. Please add the following statement, or similar, to the Methods: "This study is reported as per the Guidelines for Accurate and Transparent Health Estimates Reporting (GATHER) statement (S1 Checklist)." When completing the checklist, please use section and paragraph numbers, rather than page numbers.

* [FOR MEDIATION ANALYSES] We recommend that the study is reported according to the AGReMA statement (https://agrema-statement.org/#:~:text=AGReMA%20is%20an%20evidence%2D%20and,randomised%20trials%20and%20observational%20studies) and include the completed checklist as Supporting Information. Please add the following statement, or similar, to the Methods: "This study is reported as per the Guideline for Reporting Mediation Analyses (AGReMA) statement (S1 Checklist)." When completing the checklist, please use section and paragraph numbers, rather than page numbers.

* For all observational studies, in the manuscript text, please indicate: (1) the specific hypotheses you intended to test, (2) the analytical methods by which you planned to test them, (3) the analyses you actually performed, and (4) when reported analyses differ from those that were planned, transparent explanations for differences that affect the reliability of the study's results. If a reported analysis was performed based on an interesting but unanticipated pattern in the data, please be clear that the analysis was data driven.

* Please state in the Methods section whether the study had a prospective protocol or analysis plan. If a prospective analysis plan (from your funding proposal, IRB or other ethics committee submission, study protocol, or other planning document written before analyzing the data) was used in designing the study, please include the relevant document(s) with your revised manuscript as a Supporting Information file to be published alongside your study and cite it in the Methods section. A legend for this file should be included at the end of your manuscript. If no such document exists, please make sure that the Methods section transparently describes when analyses were planned, and when/why any data-driven changes to analyses took place. Changes in the analysis, including those made in response to peer review comments, should be identified as such in the Methods section of the paper, with rationale.

---

## [Decision Letter · Decision Letter 2]

1 Apr 2026

Dear Dr. Laisk,

I am writing on behalf of my colleague Dr. Evangelia Fourli. Thank you very much for re-submitting your manuscript "Descriptive study of prescribed hormonal contraceptive use trends in the Estonian Biobank:user profiles and a resource for genetic studies" (PMEDICINE-D-25-04320R2) for review by PLOS Medicine.

I have discussed the paper with my colleagues and the academic editor and it was also seen again by xxx reviewers. I am pleased to say that provided the remaining editorial and production issues are dealt with we are planning to accept the paper for publication in the journal.

[LINK]

We look forward to receiving the revised manuscript by Apr 08 2026 11:59PM.

Sincerely,

Alison Farrell, Ph.D.

Senior Editor

PLOS Medicine

plosmedicine.org

Comment from the Academic Editor:

Line 530:"We note that the prevalence rates...." Is prevalence the correct term?

Ed: consider usage rates? Please check use of prevalence throughout for accuracy of the term.

Requests from Editors:

Please check the grammar throughout, including the Abstract, and revise, correcting errors (e.g. lines 17, 19, 29, 34).

Please ensure that the statements in the metadata are up to date and match the statements in the manuscript files.

Please amend the COI for HC as the funding does not constitute a COI. Instead, please amend the funder statement for HC who declares funding from the Wellcome and the NIH.

Please amend the COI statement for KL to match the declaration in the questionnaire.

Please add URLs for each funder to the funding statement. The funding statement should include: specific grant numbers, initials of authors who received each award, URLs to sponsors’ websites. Also, please state whether any sponsors or funders (other than the named authors) played any role in study design, data collection and analysis, the decision to publish, or preparation of the manuscript. If they had no role in the research, include this sentence: “The funders had no role in study design, data collection and analysis, decision to publish, or preparation of the manuscript.”

We require deposition of code in a public repository and that a DOI is generated. Links to the code must be provided in the final version of the manuscript (see below).

Line 666: is consent 'informed consent'? If so, please add the qualifier.

Members of the Estonian Biobank Research Team should be identified. As they are authors, they should not be included in the Acknowledgments section but identified separately and their author contributions stated in the Author Contribution statement.

Please note that author RM has not completed the questionnaire with the COI declaration.

* All authors must declare their relevant competing interests per the PLOS policy, which can be seen here: https://journals.plos.org/plosmedicine/s/competing-interests For authors with ties to industry, please indicate whether any of the interests has a financial stake in the results of the current study.

* In the final bullet point of the Author Summary section ‘What Do These Findings Mean?’ Please include the main limitations of the study in non-technical language.

* Please confirm that your title complies with PLOS Medicine's style. Your title must be nondeclarative and not a question. It should begin with main concept if possible. "Effect of" should be used only if causality can be inferred, i.e., for an RCT. Please place the study design ("A randomized controlled trial," "A retrospective study," "A modelling study," etc.) in the subtitle (ie, after a colon). We suggest: "Prescribed hormonal contraceptive use trends and genetic associations with side effects in the Estonian Biobank: A longitudinal observational study". Please amend accordingly, retaining the style.

* Please confirm that your abstract complies with our requirements, including format (three sections: Background, Methods and Findings, and Conclusions) and providing all the information relevant to this study type https://journals.plos.org/plosmedicine/s/submission-guidelines#loc-abstract

* Please ensure that the Introduction ends with a clear description of the study question or hypothesis.

* Please ensure that all abbreviations are defined at first use throughout the text.

* Please confirm that all numbers presented in the abstract are present and identical to numbers presented in the main manuscript text.

* Abbreviations in figures need to be spelled out in the legends. VTE/ATE in figure 5, COC in figure 2.

* Please add a title for the y axis of Fig. 3, identifying the abbreviations as HC formulations.

* Please add a Box with the list of HC formulation abbreviations and their definitions and add a call out to the Box in the text.

* If statistical tests are used to generate data in figures, please identify the tests in the figure legends.

* In general we prefer that authors convert bar graphs to another format. If this is not feasible or impairs presentation, please include in the SI Tables with the data behind the bar graphs for figures 2c and 4c.

Please also note that it is difficult to discern between the blue and purple bars. Please alter the intensities for better distinction.

* Please review your text for claims of novelty or primacy (e.g. 'for the first time') and remove this language. In addition, please check that any use of statistical terms (such as trend or significant) are supported by the data, and if not please remove them.

* In the abstract, please include the important dependent variables that are adjusted for in the analyses.

* In the author summary, in the final bullet point of 'What Do These Findings Mean?', please include the main limitations of the study in non-technical language.

* Authors are responsible for providing the source code needed to replicate the study's findings in a repository (such as GitHub, SourceForge or Bitbucket) or a cloud computing service (such as Code Ocean). Protection of authors’ intellectual property will not be cause for exception. Please explain in the manuscript’s Data Availability Statement how readers can access the shared code.

* Please include the statement on code availability in the data availability statement.

* Please provide titles and legends for all figures and tables (including those in Supporting Information files). Please define all acronyms used in each figure or table in its corresponding legend.

* Please convert any stacked bar charts to another data representation for example a table, or other type of graph (see above).

* Please provide the unadjusted comparisons as well as the adjusted comparisons in all relevant Tables

* Please update the STROBE and RECORD checklists is necessary.

When completing the checklists, please use section and paragraph numbers, rather than page numbers.

* Your study is observational and therefore causality cannot be inferred. Please remove language that implies causality and refer to associations instead.

* For all observational studies, in the manuscript text, please indicate: (1) the specific hypotheses you intended to test, (2) the analytical methods by which you planned to test them, (3) the analyses you actually performed, and (4) when reported analyses differ from those that were planned, transparent explanations for differences that affect the reliability of the study's results. If a reported analysis was performed based on an interesting but unanticipated pattern in the data, please be clear that the analysis was data-driven.

Comments from Reviewers:

Reviewer #1: Thank you for submitting your revision. In reviewing the comments to the reviewers and the changes that the authors made to this publication as a result, I am happy to provide support to accept. I am happy to see that scientists are thinking about longtiduinal biobanks with contraceptive realted data. This will undoubetly support research in this area.

Reviewer #2: Thank you very much for the opportunity to review the revised manuscript. The revisions have addressed the outstanding issues, and the current version is now more clearly structured; the remaining questions have been adequately addressed. No further changes are required.

Reviewer #3: The authors have addressed all my comments.

Reviewer #4: The authors have addressed my points.

Michael Dewey

[LINK]

---

## [Editor Report · Decision Letter 3]

23 Apr 2026

Dear Dr Laisk,

On behalf of my colleagues and the Academic Editor, Isabelle Dehaene, I am pleased to inform you that we have agreed to publish your manuscript "Prescribed hormonal contraceptive use trends in the Estonian Biobank: A longitudinal observational study" (PMEDICINE-D-25-04320R3) in PLOS Medicine.

PRESS

Sincerely,

Evangelia Fourli, Ph.D.

Senior Editor

PLOS Medicine